# Depletion of Zinc Causes Osteoblast Apoptosis with Elevation of Leptin Secretion and Phosphorylation of JAK2/STAT3

**DOI:** 10.3390/nu15010077

**Published:** 2022-12-23

**Authors:** Jennifer K. Lee, Jung-Heun Ha, Do-Kyun Kim, JaeHee Kwon, Young-Eun Cho, In-Sook Kwun

**Affiliations:** 1Food Science and Human Nutrition Department, University of Florida, Gainesville, FL 32611, USA; 2Department of Food Science and Nutrition, Dankook University, Cheonan 31116, Republic of Korea; 3Zoonosis Research Institute, Jeonbuk National University, Iksan 54531, Republic of Korea; 4Department of Food and Nutrition, Andong National University, 388 Songchundong, Andong 36729, Republic of Korea

**Keywords:** low Zn, MC3T3-E1 cells, leptin, JAK2/STAT3, osteoblast apoptosis

## Abstract

Zinc (Zn) has been reported to mediate leptin secretion, and thus leptin can be an important candidate molecule linking Zn with bone formation. The present study investigated whether zinc deficiency induces leptin secretion by activating a JAK2/STAT3 signaling pathway and leads to osteoblastic apoptosis. MC3T3-E1 cells were incubated for 24 h in normal osteogenic differentiation medium (OSM) or OSM treated with either 1 μM (Low Zn) or 15 μM (High Zn) of ZnCl_2_ containing 5 μM TPEN (Zn chelator). Our results demonstrated that low Zn stimulated extracellular leptin secretion and increased mRNA and protein expression of leptin in osteoblastic MC3T3-E1 cells. The OB-Rb (long isoform of leptin receptor) expressions were also elevated in osteoblasts under depletion of Zn. Leptin-signaling proteins, JAK2 and p-JAK2 in the cytosol of low Zn osteoblast conveyed leptin signaling, which ultimately induced higher p-STAT3 expression in the nucleus. Apoptotic effects of JAK2/STAT3 pathway were shown by increased caspase-3 in low Zn osteoblasts as well as apoptotic morphological features observed by TEM. Together, these data suggest that low Zn modulates leptin secretion by activating JAK2/STAT3 signaling pathway and induces apoptosis of osteoblastic MC3T3-E1 cells.

## 1. Introduction

Bones serve vital functions in our body by providing motility, support, protection of internal organs, and storage for minerals like calcium and phosphate [1]. Bone metabolism involves major nutrients including calcium, phosphorus and magnesium, and potassiumas well as trace metals like as B, Se, Zn, Fe, and Cu [2]. Bone homeostasis is maintained by coordinated remodeling processes by key bone cells: osteoclasts, osteoblasts, osteocytes, and bone lining cells [3]. Osteoblasts are derived from Mesenchymal stem cell (MSCs) that originate in bone marrow and osteoclasts are multinucleated cells differentiated from hematopoietic stem cells (HSCs) [4]. 

Zinc (Zn) is an essential trace mineral associated with various biological roles including bone homeostasis, appetite control, fat metabolism, insulin resistance and obesity, and immunity. It is widely known that Zn is involved in critical bone functions ranging from bone formation and development to maintenance of healthy bones in humans and other mammals [5,6]. Being present at a concentration of up to 300 mg/g in bone [7], Zn has been well demonstrated to stimulate osteoblastic activity and collagen synthesis by inhibiting osteoclastic bone resorption [8,9]. Zn depletion markedly reduced alkaline phosphatase (ALP) activity and the concentration of Ca, Mg and P in rats [10]. Low Zn has been shown to negatively impact bone formation, differentiation, and mineralization in vivo and in vitro [11,12]. Geiser et al. demonstrated that conditional knockout of intestinal Zn transporter ZIP4 in mice has recapitulated symptoms associated with acrodermatitis enteropathica and resulted in a rapid loss of bone and muscle mass and body weight [13]. In contrast, Zn supplementation in children has been shown to stimulate both skeletal growth and maturation [14].

Leptin, the product of the ob gene, is a hormone belonging to the cytokine family that is primarily synthesized and secreted by adipocytes [15,16]. Leptin is widely known for its critical role in regulation of energy homoeostatis and expenditure [17]. The biological actions of leptin on target tissues are mediated through its specific receptors, OB-Rs [18]. Among them, the long signal-transducing form of leptin receptor, OB-Rb, is the most important signal-transducing unit [19] and expressed in the osteoblasts [20]. Interestingly, emerging evidence has demonstrated that leptin is also associated with neuroendocrine regulation and bone metabolism [21,22]. There are reports on bone cell types producing substantial amounts of leptin which includes skeletal muscle, bone marrow, primary cultures of [23] as well as cultures of MC3T3-E1 cells [24]. Leptin originates in adipose tissue and acts directly on osteoblasts to increase bone growth [25]. This suggests that depletion of Zn might be a mediator of leptin secretion impacting reduction of osteoblast differentiation. However, the mechanism by which high leptin secretion affects osteoblast apoptosis under low Zn is not clearly elucidated.

The primary objective of this study was to investigate whether low Zn modulates leptin and long-form leptin receptor (OB-Rb) protein expression through the JAK2/STAT3 leptin signaling pathway, thus inducing osteoblast apoptosis in osteoblastic MC3T3-E1 cells.

## 2. Materials and Methods

### 2.1. Cell Culture and Zn Treatment

MC3T3-E1 subclone 4 (SC4, high osteoblast differentiation, ATCC, CRL-2593) murine pre-osteoblast was purchased from ATCC Cell Bank (Manassas, VA, USA). To test the effects of Zn on osteoblasts, the of 1 × 10^5^ of MC3T3-E1 cells were maintained in Minimum Essential Medium α (α-MEM) containing 10% FBS, 1 mM sodium pyruvate and 1% penicillin and streptomycin. The cells were incubated under a humidified atmosphere at 37 °C and 5% CO_2_ to reach up to 80% confluency. Subsequently, the cells were treated with either 1 μM (low Zn) or 15 μM (high Zn) of ZnCl_2_ (Sigma, St. Louis, MO, USA) along with 5 μM of intracellular Zn chelator N,N,N′,N′-Tetrakis(2-pyridylmethyl)ethylenediamine, TPEN; Sigma) to induce the cellular Zn depletion. The normal osteogenic differentiation medium (OSM; growth medium supplemented with osteogenic growth factors of 3 mM glycerol-2-phosphate and 50 μg/mL ascorbic acid) without Zn and TPEN treatment was used as a normal control. MC3T3-E1 cells were then exposed to OSM, low Zn, or high Zn for 24 h.

### 2.2. Cell Viability

Cell viability was assessed using the 3-(4,5-dimethylthiazol-2-yl)-2,5-diphenyltetrazolium bromide (MTT) colorimetric assay (Sigma, St. Louis, MO, USA), as previously described [26]. Briefly, MC3T3-E1 cells were seeded into a 96-well microplate (SPL Life Science, Seongnam, Republic of Korea) at a density of 1 × 10^4^ cells/well in 96 well plates and incubated under 37 °C and 5% CO_2_ conditions. After 24 h incubation, the cells were treated with different Zn concentrations (0, 0.25, 0.5, 1, 3, or 15 μM) for additional 24 h. Subsequently, the MTT reagent was added to react for 3 h. The culture medium in each well was then removed and dissolved in filtered dimethyl sulfoxide (DMSO). The absorbance was determined at 570 nm using the microplate reader (TECAN, Männedorf, Switzerland).

### 2.3. Alkaline Phosphatase (ALP) Activity Assay

ALP activity was measured as previously described [9,12,26]. Briefly, ALP activity in cellular cytosolic supernatant or culture medium was measured based on the conversion of p-nitrophenyl-phosphate (*p*NPP) to p-nitrophenol (pPNP). The absorbance was measured at 450 nm, and the obtained values were normalized with respect to the protein concentration measured by the BCA protein assay (Thermo Fisher Scientific, Rockford, IL, USA). The ALP activity within cells and culture medium is exhibited as Unit (nmol pNP)/mg protein/min and Unit (nmol pNP)/mL/min, respectively.

### 2.4. Leptin Quntification

The release of leptin into the culture medium was determined by a quantitative sandwich enzyme-linked immunosorbent assay (ELISA) according to the manufacturer’s instructions (R&D Systems, Minneapolis, MN, USA). 

### 2.5. Transmission Electron Microscopy

Cells (*n* = 5/each sample) were fixed in 2.5% glutaraldehyde and 1% osmium tetroxide, respectively, followed by a dehydration in graded acetone of 50%, 70%, 90% and 100%. Fixed cells were paraffin-embedded and cut at 50–100 nm thickness. The sections were then stained with 3% uranyl acetate and lead nitrate. All specimens were examined with the transmission electron microscope (HT-7700 TEM; Hitachi, Tokyo, Japan). Apoptosis was characterized based on the cell organelle features. 

### 2.6. Quantitative Real-Time qPCR Analysis and RT-PCR Analysis

Quantitative real-time PCR (RT-qPCR) was performed to quantify transcription levels of Leptin, OB-Rb, Bax, and Caspase-3as previously described [9,12,26]. Briefly, total RNA was isolated using Trizol (Thermo Fisher Scientific), followed by an overnight precipitation and RNase-free DNase I treatment. Subsequently, 600 ng of RNA was reverse-transcribed into cDNA using the cDNA reverse-transcription kit (Thermo Fisher Scientific). 

To detect Leptin and OB-Rb mRNA expression, the PCR products were separated on 1·2% agarose gels, stained with ethidium bromide. At the 30th cycle, PCR products were clearly visible on an agarose gel and photographed under UV light using a digital camera for band intensity quantification.

To detect relative Caspase-3 and Bax mRNA expression, SYBR-Green RT-qPCR was completed using a Mx3000p System (Thermo Fisher Scientific). The expression of experimental genes was normalized to a stable reference gene, glyceraldehyde 3-phosphate dehydrogenase (GAPDH). The delta CT (ΔCt) method was used to calculate relative quantification of mRNA expression. 

Sequences of forward and reverse primers are listed in Table 1.

### 2.7. Western Blotting Analysis

Protein expression of ALP, Pro COL-1, Leptin, OB-Rb, JAK2, p-JAK2, STAT3, p-STAT3, cleaved (c)-Caspase 3, and cleaved (c)-PARP-1 was assessed using western blotting, as previously described [9,12,26]. Proteins were extracted by the NE-PER™ Nuclear and Cytoplasmic Extraction Reagents kit (ThermoFisher Scientific) for separating nuclear and cytoplasmic proteins. Protein concentration was quantified using the Pierce BCA Protein Assay Kit (Thermo Fisher Scientific. Equal amounts of protein from each sample were separated by sodium dodecyl-sulfate polyacrylamide gel electrophoresis (SDS/PAGE) and electrophoretically transferred onto nitrocellulose membranes. After blocking with 5% skim milk in 1X PBS with 0.1% Tween-20, the membranes were incubated with primary antibodies (target proteins) for overnight at 4 °C followed by another incubation with secondary antibodies. Blots were imaged by using HRP-conjugated secondary antibodies combined with ECL substrates (Thermo Fisher Scientific). The band intensities were quantified using ImageJ software (National Institute of Health, MD, USA). Primary and secondary antibodies used in the experiments are summarized in Table 2.

### 2.8. Fluorescent Microscopy

For detecting c-Caspase-3 using immunofluorescence, MC3T3-E1 cells were initially plated onto chamber slides. After removal of the blocking solution, the cells were incubated with the specific antibody, as indicated, at 4 °C overnight. Anti-c-Caspase-3 antibody were used at 1:300 dilution. For immunofluorescence detection, the cells were incubated with Alexa Fluor 488-labeled anti-rabbit secondary antibody (Invitrogen). For nuclear staining, the cells were incubated with 1 mg/mL 4′,6′-diamino-2-phenylindole (DAPI) for 5 min. The cells were fixed 2.5% formalin and stained with Hoechst staining. Fluorescence images were collected by using fluorescent microscope (ThermoFisher Scientific). 

### 2.9. Statistical Analysis

The experimental data were expressed as the mean ± SEM. The mean differences between groups were considered significant when *p* < 0.05. Statistical analysis of the data was performed by one-way ANOVA using SPSS 27 program (SPSS Inc., Chicago, IL, USA). Once significance was noted, Tukey’s honestly significant difference (HSD) method was further applied to find differences between individual groups [9,10,12,26].

## 3. Results

### 3.1. Low Zn Decreased Cell Viability, ALP Activity, and Bone-Related Protein Expression in MC3T3-E1 Cells

Zn is a vital trace mineral required for cell viability, DNA repair, and apoptosis [27]. In order to better understand the effects of Zn levels on functional integrity of osteoblasts, MC3T3-E1 cells were used in this study. Cell viability was assessed using the MTT assay after treated with different concentrations of Zn (0, 0.25, 0.5, 1, 3, or 15 μM) and 5 μM TPEN for 24 h. OSM is the normal osteogenic differentiation medium without Zn chelator, TPEN. Cell viability showed a significant dose-related increase (*p* < 0.05) as Zn concentrations increased (Figure 1A). Low Zn appeared to show reduced cell viability. In this study, either 1 μM (low Zn) or 15 μM (high Zn) of Zn with TPEN were treated to the cells for further analyses. This ensures that low Zn conditions were present in this study. Alkaline phosphatase (ALP) is a robust biomarker for osteoblastic activity during early osteoblast differentiation [12]. The effects of low Zn on MC3T3-E1 cell differentiation were measured by cellular and medium ALP activity (Figure 1B). The cellular ALP activity of MC3T3-E1 cells was not significantly decreased by low Zn Figure 1B upper panel). However, high Zn showed elevated cellular ALP activity may indicate active bone formation The ALP activity of MC3T3-E1 cell culture medium was not significantly changed according to Zn content (Figure 1B lower panel). In addition, protein expression of bone formation biomarkers such as ALP and pro collagen type I (Pro COLI) was assessed. As shown in cellular ALP activity, protein expression of ALP and Pro COLI was increased in high Zn cells, as compared to OSM (Figure 1C,D). No significant difference of ALP protein expression was noted between low Zn and OSM. These results may indicate that low Zn in osteoblasts is associated with decreased cell viability, cellular ALP level, and bone-related protein expression.

### 3.2. Low Zn Increased Leptin Secretion and Leptin mRNA and Protein Expression in MC3T3-E1 Cells

To evaluate the secretion of leptin in MC3T3-E1 cells under various Zn contents, we measured leptin concentrations in the medium using ELISA analysis. The leptin secretion was higher (*p* < 0.05) in low Zn supplementation levels (0–1 μM), as compared to OSM and high Zn concentrations (3–15 μM) in MC3T3-E1 cells (Figure 2A). Using RT-qPCR and western blotting, we assessed leptin mRNA and protein expression in MC3T3-E1 cells as a result of Zn deficiency. As demonstrated earlier in Figure 2A, the leptin mRNA and protein expression in MC3T3-E1 cells was also significantly increased in low Zn (Figure 2B,C). These results suggest that the leptin secretion, mRNA and protein expression were regulated by Zn levels, indicating the importance of Zn modulation of leptin signaling in osteoblasts.

### 3.3. Leptin Receptor (OB-Rb) mRNA and Protein Expression by Low Zn in MC3T3-E1 Cells

As a negative feedback regulator of energy homeostasis, leptin binds to its transmembrane receptor (OB-Rb) which further signals to activate Janus kinase/signal transducer and activator of transcription (JAK-STAT) pathway [28]. To confirm the expression of leptin receptors by low Zn, we examined mRNA and protein expression of the leptin receptor in MC3T3-E1 cells exposed with different Zn repletion levels (either 1 or 15 μM) using RT-qPCR and western blotting, respectively. The mRNA expression of a long form of the leptin receptor (OB-Rb) in MC3T3-E1 cells was significantly increased by Zn deficiency when assessed by RT-qPCR analysis (Figure 3A). However, a short isoform (OB-Ra) was not detected in MC3T3-E1 cells. The western blotting also showed the long form of the leptin receptor (OB-Rb) protein was elevated in low Zn compared to OSM and high Zn in MC3T3-E1 cells (Figure 3B). These results indicate that low Zn modulate the long form of the leptin receptor (OB-Rb) as well as leptin production in MC3T3-E1 cells.

### 3.4. JAK/p-STAT Protein Expression by Low Zn in MC3T3-E1 Cells

JAK/STAT pathway plays critical roles in delivering of signals (induced by growth factors and cytokines) from cell membrane-bound receptors to the nucleus to allow transcription of target genes [28]. As a major regulator of energy homeostasis and food intake, leptin binds to its receptor OB-R which subsequently activates JAK-STAT signaling cascade. To further determine the effects of Zn on leptin-signaling proteins such as JAK2 and p-STAT3, the western blotting was performed. First, we confirmed the cytoplasmic and nucleus marker proteins to validate extraction method. Our results showed that the JAK2 and phosphorylated JAK2 (p-JAK2) protein in MC3T3-E1 cells trended higher by low Zn (Figure 4A). Additionally, the phosphorylated-STAT3 (p-STAT3) protein was significantly increased by low Zn compared to OSM or high Zn (Figure 4B). Therefore, these results may imply that leptin signaling through JAK2/STAT3 pathway was stimulated by low Zn in MC3T3-E1 cells.

### 3.5. Low Zn Increased Apoptosis Signals in MC3T3-E1 Cells

Depletion of Zn resulting osteoporosis through the loss of bone mass has been extensively studied [5,6,8,9,12]. To confirm whether low Zn induced osteoblastic cell apoptosis, we visualized several significant hallmark events of apoptosis using transmission electron microscopy (TEM) microscope. Remarkably, TEM image revealed the chromatin condensation at the margins of nuclei, cell shrinkage, membrane blebbing, and mitochondrial vacuolization in MC3T3-E1 cells supplemented with 1 μM of Zn (low Zn) (Figure 5).

To investigate the apoptosis-inducing effect of Zn, MC3T3-E1 cells were treated with low Zn. Cells were analyzed using fluorescent microscopy with Hoechst staining. As shown in Figure 6, chromatin condensation, nuclear fragmentation and apoptotic bodies were observed in the treated cells. Additionally, cleaved-caspase3 were elevated in low Zn compared to OSM or high Zn (Figure 6). The results revealed that low Zn showed the apoptosis modulation.

We have also assessed the mRNA and protein expression of apoptosis-related markers. The results showed that the expression of apoptosis-related proteins such as c-Caspase3, c-PARP1, and Bax was elevated in MC3T3-E1 cells under low Zn condition (Figure 7A). We found that the Caspase 3 and Bax mRNA expression was increased in MC3T3-E1 cells by low Zn (Figure 7B). These results suggest that leptin signaling through JAK2/STAT3 pathway could be induced by the osteoblastic cell apoptosis under the low Zn condition.

## 4. Discussion

Bone is a dynamic tissue that undergoes continuous remodeling process regulated by various local and systemic regulators [29]. The major systemic regulators include hormones such as parathyroid hormone (PTH), calcitonin (CT), vitamin D3 [1,25(OH)2 vitamin D3], and estrogen as well as factors such as insulin-like growth factors (IGFs), fibroblast growth factors (FGFs), tumor growth factor-beta (TGF-β), epidermal growth factors (EGFs), bone morphogenetic proteins (BMP), and cytokines [30]. The local regulation of bone remodeling is associated with cytokines like TNF-α, IL-6, and IL-10 and parathyroid hormone-related protein (PTHrP) [30]. Together, these factors maintain skeletal homeostasis by functioning to activate or repress osteogenic transcription as well as hormonal controls [31]. Our previous studies suggested that low Zn disrupts osteoblast differentiation and mineralization in cell and mouse models [9,10,12,26]. Remarkably, the pleotropic hormone leptin has now been reported to modulate bone mineral density (BMD) or bone turnover as a key element [23]. In this study, we demonstrated that low Zn in osteoblasts promoted leptin secretion and decreased osteoblast differentiation. In response to leptin stimulation in the extracellular matrix of Zn deprived cells, the mRNA and protein expression of leptin receptor (OB-Rb) was increased via the activation of JAK2/STAT3 signaling pathway via osteoblast apoptosis.

Leptin is a cytokine-like hormone secreted by adipocytes which plays important roles in regulating body weight, metabolism and reproductive function [15,16]. Leptin acts as an afferent signal in a negative feedback loop predominantly in the hypothalamus to regulate food intake, energy expenditure, and fat mass [32]. Zn plays a significant role in appetite control [33]. It has been reported that serum leptin and leptin mRNA levels in inguinal adipocytes were increased due to low Zn followed by reduced food intake in animals [34]. In contrast, some studies have demonstrated that low Zn reduced leptin concentration in human [35] and rodents [36]. In the osteoblasts and adipocytes as well as chondroblasts and myoblasts may have differentiated from bone marrow stromal cells, all of these cells might respond to the same leptin action in vitro models [23,24]. In present study, leptin levels in the culture media as well as leptin gene and protein expression were all increased in MC3T3-E1 cells under low Zn and decreased osteoblast differentiation.

Leptin acts through its transmembrane receptor (OB-Rb), which is a long, fully active isoform of the leptin receptor (OB-R) [19]. It has been well established that binding of leptin to OB-Rb leads to JAK2 activation which subsequently phosphorylates tyrosine residues [36]. As a JAK2 downstream mediator, STAT3 is mainly induced by tyrosine phosphorylation [28]. The phosphorylated-STAT3 is then delivered to the nucleus to bind STAT3-responsive element on the specific promotor region to induce gene expression [37,38]. Indeed, it was reported that STAT3 can be activated by leptin in the hypothalamus [39]. Several studies have demonstrated that OB-Rb gene was expressed in rat osteoblasts [40] and human osteoblasts [41], although Ducy et al., did not observe any OB-R in primary mouse osteoblasts [23]. Our data showed that OB-Rb gene was expressed in MC3T3-E1 cells and increased its gene expression under conditions of low Zn. Our data shown that osteoblastic MC3T3-E1 cells under low Zn expressed higher both OB-Rb and leptin mRNA/protein suggesting an increased the leptin secretion. Furthermore, our results suggested p-JAK2 and p-STAT3 protein was increased in MC3T3-E1 cells in low Zn. Therefore, low Zn may possibly modulate the JAK2/p-STAT3 signals through leptin receptor (OB-Rb) in MC3T3-E1 cells.

Apoptosis in osteoblast has been shown to contribute to reduced bone mass and osteoporosis [42]. Cell apoptosis can be induced by a cross-talk between the mitochondria-mediated pathway and cell death receptors [43]. Activated STAT3 (i.e., p-STAT3) promotes the upregulation of various genes associated with cell survival and apoptosis [37,38]. It has been reported that the transcript levels of STAT3-regulated anti-apoptotic genes Bcl-2 and MCL-1 or proapoptotic gene such as caspase-3 were both upregulated in Chronic lymphocytic leukemia (CLL) cells from patients [44] and myeloid cells [45]. In fact, caspase-3 is the most important effector caspase of the apoptotic response network [46]. Following an apoptotic stimulus, the mitochondrial (intrinsic) apoptosis pathway is mediated by the Bcl-2 family and their membrane interactions [47]. Within Bax is a pro-apoptotic protein, whereas Bcl-2 is an anti-apoptotic protein. An imbalance of Bax and Bcl-2 proteins may lead to the loss of the mitochondrial membrane potential (MMP) and cytochrome c release, activating caspase-3 apoptotic cascade [48]. It has been reported that low Zn induces apoptosis through this aforementioned mitochondria-mediated pathway in osteoblastic cells [43]. Our data clearly showed that low Zn in osteoblasts increased the c-Caspase3 and apoptotic morphological features shown by Hoechst staining and TEM image. Furthermore, gene and protein expression of c-Caspase-3 and Bax were elevated in low Zn condition. In the current investigation, leptin and the OB-Rb were both elevated in osteoblasts under Zn depletion. These findings imply that low Zn may modulate leptin secretion by activating JAK2/STAT3 signaling pathway, as inducing osteoblast apoptosis although future in vivo experiments will be required to confirm this explanation.

## Figures and Tables

**Figure 1 nutrients-15-00077-f001:**
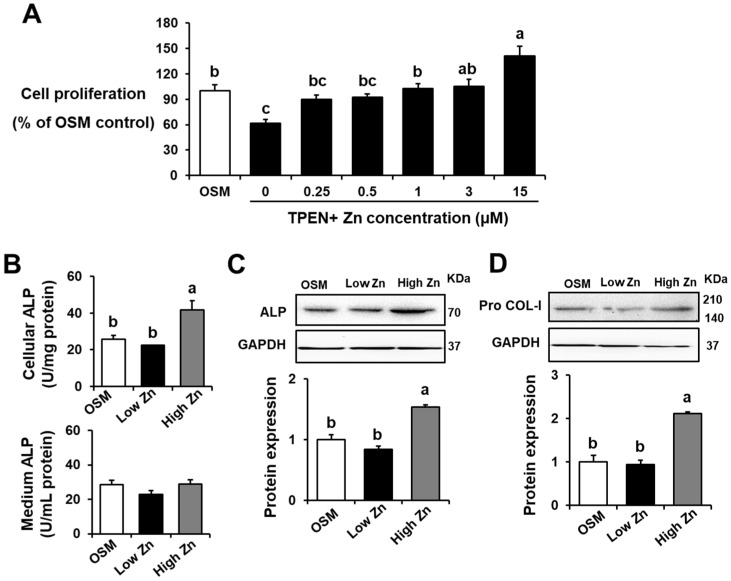
Effects of low Zn on cell viability, alkaline phosphatase (ALP) activity, osteogenesis-related protein expression in MC3T3-E1 cells. OSM is the normal osteogenic differentiation medium without Zn chelator, 5 μM TPEN. 1 μM (low Zn) and 15 μM (high Zn) are the media Zn added as designated with 5 μM TPEN. (**A**) Cell viability of MC3T3-E1 cells cultured with various concentrations of Zn (0, 0.25, 0.5, 1, 3, and 15 uM) with TPEN was measured by the MTT assay. The results are shown as % of OSM. (**B**) ALP activity of culture media or MC3T3-E1 cells treated with either low or high Zn was measured as a marker for osteoblast maturation. One unit (U) of ALP activity refers to a nmol *p*-nitrophenol/mg of protein/min in cells and a nmol *p*-nitrophenol/mL/min in media. (**C**,**D**) The protein expression of osteogenesis-related proteins, ALP and Pro COL-1, was analyzed by western blotting. GAPDH was utilized as loading control for protein normalization. Data represent means ± SEM. One-way ANOVA model was used to analyze the dataset. Once significance is noted, Tukey’s HSD method was further applied to find significances between groups. Groups with different superscript letters are significantly different from one another, *p* < 0.05.

**Figure 2 nutrients-15-00077-f002:**
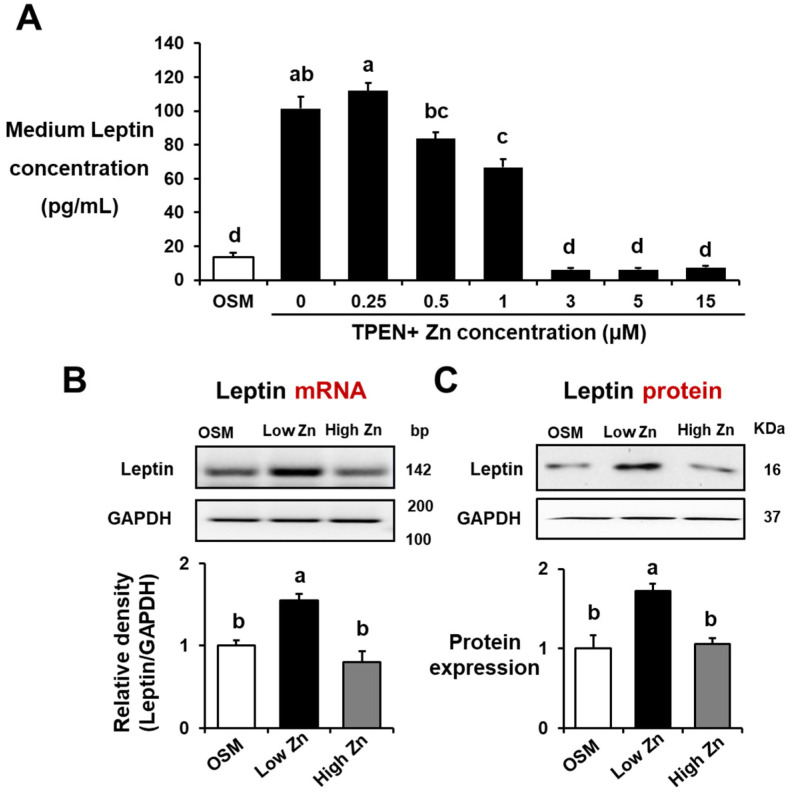
Effects of low Zn on secretion and mRNA and protein expression of leptin in MC3T3-E1 cells. OSM is the normal osteogenic differentiation medium without Zn chelator, 5 μM TPEN. 1 μM (low Zn) and 15 μM (high Zn) are the media Zn added as designated with 5 μM TPEN. (**A**) Leptin secretion of MC3T3-E1 cells treated with different Zn doses (0, 0.25, 0.5, 1, 3, and 15 uM) with TPEN was measured using ELISA. (**B**,**C**) mRNA and protein expression of leptin in MC3T3-E1 cells treated with either low Zn or 15 high Zn were analyzed by RT-qPCR and western blotting. GAPDH was utilized as loading control for protein normalization. Data represent means ± SEM. One-way ANOVA model was used to analyze the dataset. Once significance is noted, Tukey’s HSD method was further applied to find significances between groups. Groups with different superscript letters are significantly different from one another, *p* < 0.05.

**Figure 3 nutrients-15-00077-f003:**
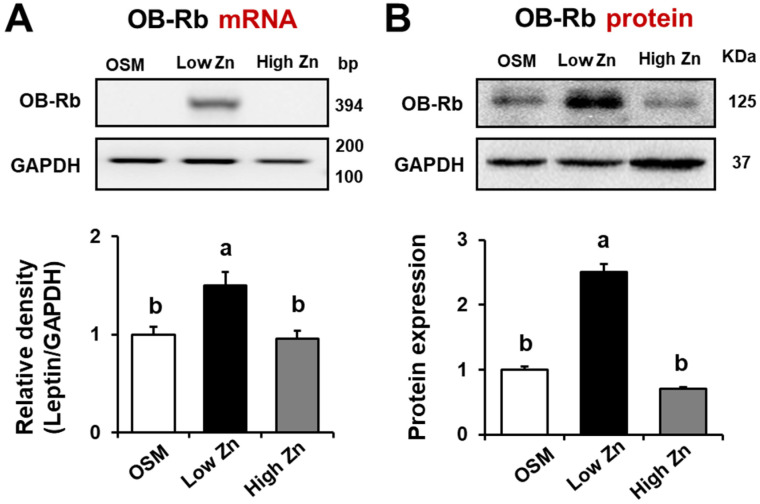
Effects of low Zn on mRNA and protein expression of the OB-Rb in MC3T3-E1 cells. OSM is the normal osteogenic differentiation medium without Zn chelator, 5 μM TPEN. 1 μM (low Zn) and 15 μM (high Zn) are the media Zn added as designated with5 μM TPEN. (**A**,**B**) mRNA and protein expression of the long form leptin receptor (OB-Rb) in MC3T3-E1 cells treated with either low or high Zn were analyzed by RT-qPCR and western blotting. GAPDH was utilized as loading control for protein normalization. Data represent means ± SEM. One-way ANOVA model was used to analyze the dataset. Once significance is noted, Tukey’s HSD method was further applied to find significances between groups. Groups with different superscript letters are significantly different from one another, *p* < 0.05.

**Figure 4 nutrients-15-00077-f004:**
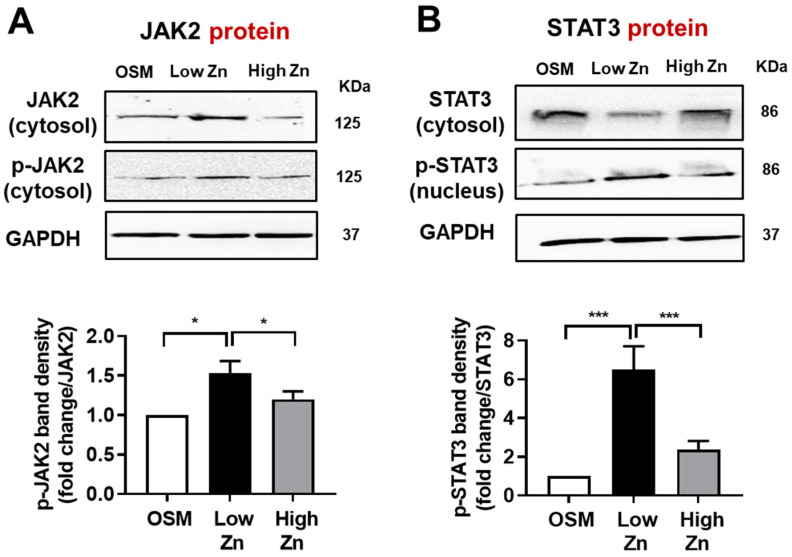
Effects of low Zn on protein expression of JAK2, p-JAK2, STAT3, p-STAT3 in MC3T3-E1 cells. (**A**,**B**) Protein expression of JAK2, p-JAK2, STAT3, and p-STAT3 in MC3T3-E1 cells treated with either low or high Z nwere analyzed by Western blotting. GAPDH was utilized as loading control for protein normalization. Data represent means ± SEM. The statistical significance between values for each group was assessed by Dunnett’s *t*-test. *** *p* < 0.001 between OSM, high Zn, and low Zn groups; * *p* < 0.05 between OSM, high Zn, and low Zn groups.

**Figure 5 nutrients-15-00077-f005:**
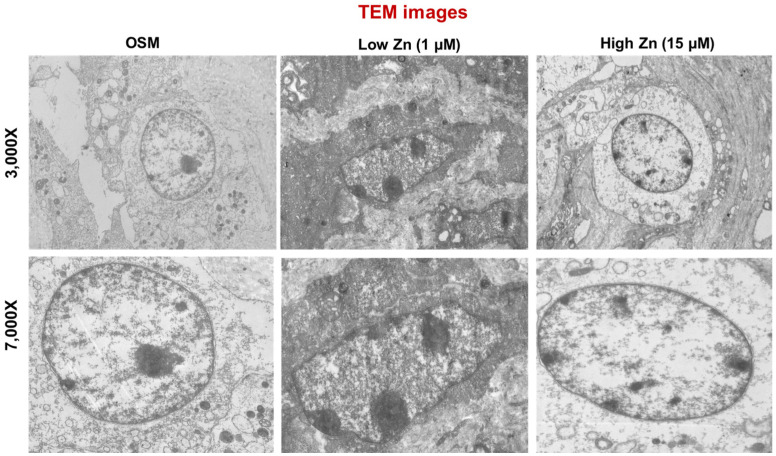
Apoptotic effects of Zn deficiency in MC3T3-E1 cells. OSM is the normal osteogenic differentiation medium without Zn chelator, TPEN. 1 μM (low Zn) and 15 μM (high Zn) are the media Zn added as designated with TPEN. Representative transmission electron microcopy (TEM) images (3000Xor 7000X) of MC3T3-E1 cells (*n* = 5) treated with either low or high Zn were visualized. The normal OSM serves as controls.

**Figure 6 nutrients-15-00077-f006:**
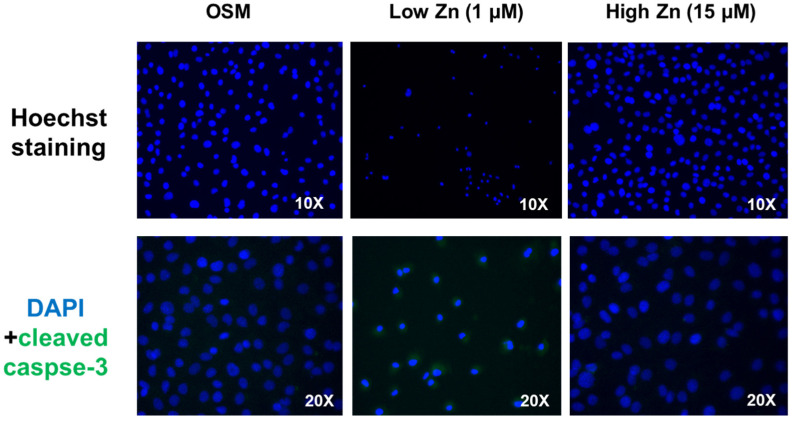
Osteoblast apoptosis observed using Hoechst staining and c-Caspase-3. MC3T3-E1 cells were treated with low or high Zn for 24 h. The apoptotic cells exhibited chromatin condensation, nuclear fragmentation and apoptotic bodies.

**Figure 7 nutrients-15-00077-f007:**
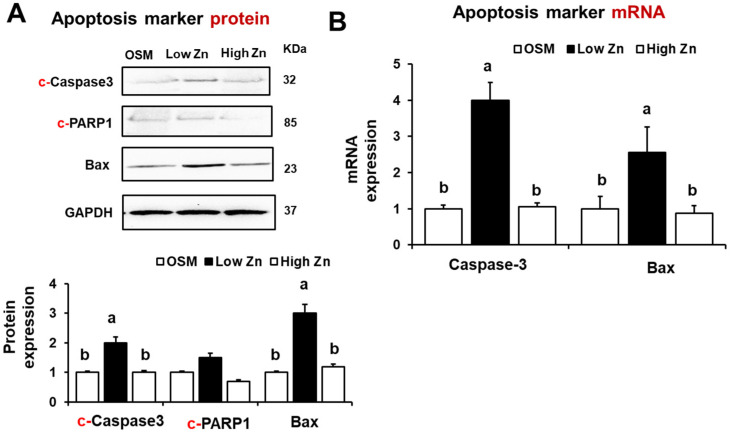
Apoptosis marker proteins induced by low Zn in MC3T3-E1 cells. OSM is the normal osteogenic differentiation medium without Zn chelator, 5 μM TPEN. 1 μM (low Zn) and 15 μM (high Zn) are the media Zn added as designated with TPEN. (**A**) Protein expression of c(cleaved)-Caspase3, Bax, c(cleaved)-PARP1 in MC3T3-E1 cells were analyzed by western blotting. GAPDH was utilized as loading control for protein normalization. (**B**) mRNA expression of Caspase-3 or Bax was quantified by RT-qPCR. Data represent means ± SEM. One-way ANOVA model was used to analyze the dataset. Once significance is noted, Tukey’s HSD method was further applied to find significances between groups. Groups with different superscript letters are significantly different from one another, *p* < 0.05.

**Table 1 nutrients-15-00077-t001:** PCR primers used in this study.

Transcript	Forward Primer	Reverse Primer
Leptin	GAG ACC CCT GTG TCG GTT C	CTG CGT GTG TGA AAT GTC ATT G
OB-Rb	GGG TAA TAC TTA AAC AGT GAC C	CTA TCT GAA AAT AAA AAC TTC ATG
Caspase-3	TGG TGA TGA AGG GGT CAT TTA TG	TTC GGC TTT CCA GTC AGA CTC
Bax	CTA CAG GGT TTC ATC CAG	CCA GTT CAT CTC CAA TTC G
GAPDH	TCC ACT CAC GGC AAA TTC AAC	TAG ACT CCA CGA CAT ACT CAG C

OB-Rb, the long isoform of the leptin receptor; GAPDH, glyceraldehyde 3-phosphate dehydrogenase.

**Table 2 nutrients-15-00077-t002:** Antibodies for western blotting analysis.

	Antibody	Dilution Factor	Corporation	Cat. No.
Primary antibody	ALP	1:1000	Santa Cruz	sc-271431
	Pro COL-I	1:1000	Santa Cruz	sc-166572
	Leptin	1:1000	Santa Cruz	sc-471278
	OB-R	1:1000	Santa Cruz	sc-8391
	JAK2	1:1000	Cell signaling	#3230
	p-JAK2	1:1000	Cell signaling	#3774
	STAT3	1:1000	Cell signaling	#9139
	p-STAT3	1:1000	Cell signaling	#52075
	c-Caspase-3	1:1000	Santa Cruz	sc-7272
	Bax	1:1000	Santa Cruz	Sc-7480
	c-PARP1	1:1000	Santa Cruz	sc-8007
	GAPDH	1:1000	Cell signaling	#2118
Secondary antibody	Goat anti-mouse-HRP	1:5000	Santa Cruz	sc-516102
	Goat anti-rabbit-HRP	1:5000	Santa Cruz	sc-2357

ALP, alkaline phosphatase; Pro COL-I, Procollagen type I; OB-R, leptin receptor short and long isoform; JAK2, Janus kinases 2; p, phosphorylation; STAT3, signal transducer and activator of transcription 3.

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
