# Peer review of "Depletion of Zinc Causes Osteoblast Apoptosis with Elevation of Leptin Secretion and Phosphorylation of JAK2/STAT3"

_nutrients, 2022, doi:10.3390/nu15010077_

Round 1
Reviewer 1 Report
Overall, I felt that this was an interesting study. This study suggested that leptin secreted by Zn-deficient osteoblastic MC3T3-E1 cells induces apoptosis through an autocrine signaling pathway that activates its own JAK2/STAT3 signaling pathway.
The results of the experiment and the discussions derived from them are generally appropriate. I have some major and minor comments listed below to be addressed.
Major
The primer set used for gene expression analysis of Bcl-XL is inappropriate (primer blast did not hit the appropriate amplicon). The primer sets used in this experiment have completely different Tm values for each gene. For example, the primer sets used for OB-Rb and GAPDH differ in Tm value by approximately 10°C, making it difficult to perform relative quantification using the DDCt method under the same PCR conditions.
RIPA buffer contains detergents (SDS and SDC) that destroy the nuclear membrane and solubilize nuclear proteins, so it is not suitable for lysate adjustment when separating nuclear and cytoplasmic proteins. GAPDH is unsuitable as an internal standard for nuclear proteins.
The figure legend should also show that TPEN was used to chelate zinc, except in the OSM group.
The novelty of this study could be enhanced if you could show that the activation of JAK3/STAT3 signaling and apoptosis induced by zinc depletion were suppressed in leptin-knockdown osteoblasts.
Minor
Line 74
Please indicate the well plate or dish used.
Line 88
Please provide proper citations.
Line 89
Please indicate the appropriate number of cells.
Line 102
pNP should be corrected to pPNP.
Line 128
In addition to the manufacturer name, the part number should be added to the table summarizing the antibody list.
Line 164-166
For the convenience of the reader, these results should be considered due to zinc depletion due to the addition of TPEN.
Line 177
Poor consistency (were and was) with the legend in Figure 1.
Line 180-182
This observation is inconsistent with the results of the previous article.
Line 192-193
Graph results are not shown in %.
Line 203
RT-PCR should be corrected to qRT-PCR.
Line 211-212
Did you use a chelating agent (TPEN) in the experiments in this figure?
Line 222-224 and 249-250
Please provide appropriate citations (used in the discussion).
Line 280-281
Is there a significant difference in Bcl-2 gene expression levels between Zn- and Zn+?
354-355
These are not discussion, but the results of Fig.4B.
Line 365-366
Why do we need a line break here?
Author Response
Overall, I felt that this was an interesting study. This study suggested that leptin secreted by Zn-deficient osteoblastic MC3T3-E1 cells induces apoptosis through an autocrine signaling pathway that activates its own JAK2/STAT3 signaling pathway.
The results of the experiment and the discussions derived from them are generally appropriate. I have some major and minor comments listed below to be addressed.
Major
The primer set used for gene expression analysis of Bcl-XL is inappropriate (primer blast did not hit the appropriate amplicon). The primer sets used in this experiment have completely different Tm values for each gene. For example, the primer sets used for OB-Rb and GAPDH differ in Tm value by approximately 10°C, making it difficult to perform relative quantification using the DDCt method under the same PCR conditions.
Response: We appreciate the reviewer’s expert points. Therefore, we deleted the Bcl-xL gene data and added the protein expression of Bax and PARP.
RIPA buffer contains detergents (SDS and SDC) that destroy the nuclear membrane and solubilize nuclear proteins, so it is not suitable for lysate adjustment when separating nuclear and cytoplasmic proteins. GAPDH is unsuitable as an internal standard for nuclear proteins.
Response: We appreciated your excellent comment. We used the NE-PER™ Nuclear and Cytoplasmic Extraction Reagents kit for separating nuclear and cytoplasmic proteins. Therefore, we modified material method and added internal standard for nuclear proteins in Figure 4.
The figure legend should also show that TPEN was used to chelate zinc, except in the OSM group.
Response: We fully understand and agree with the reviewer’s comments. Therefore, we modified the figure legends. OSM is the ‘normal osteogenic differentiation medium’ without Zn chelator, TPEN. 1 μM (low Zn) and 15 μM (high Zn) are the media Zn added as designated with TPEN. Revised on the manuscript.
The novelty of this study could be enhanced if you could show that the activation of JAK3/STAT3 signaling and apoptosis induced by zinc depletion were suppressed in leptin-knockdown osteoblasts.
Response: We appreciated the reviewer's comment. Honestly, we are not available to handle these experiments. Once again, thank you very much for your comments.
Minor
Line 74, Please indicate the well plate or dish used.
Response: We appreciated the reviewer's comment. Therefore, added the well plates.
Line 88, Please provide proper citations.
Response: We appreciated the reviewer's comment. Therefore, checked the proper citations.
Line 89, Please indicate the appropriate number of cells.
Response: We appreciated the reviewer's comment. Therefore, added the number of cells.
Line 102, pNP should be corrected to pPNP.
Response: We appreciated the reviewer's comment. Therefore, changed it as below. p-nitrophenyl-phosphate (pNPP) to p-nitrophenol (pPNP ).
Line 128, In addition to the manufacturer name, the part number should be added to the table summarizing the antibody list.
Response: We appreciated the reviewer's comment. Therefore, modified the antibody list.
Line 164-166, For the convenience of the reader, these results should be considered due to zinc depletion due to the addition of TPEN.
Response: We appreciated the reviewer's comment. Therefore, modified the results.
Line 177, Poor consistency (were and was) with the legend in Figure 1.
Response: We appreciated the reviewer's comment. Therefore, modified the legend in Figure 1.
Line 180-182, This observation is inconsistent with the results of the previous article.
Response: We appreciated the reviewer's comment. Therefore, changed it.
Line 192-193, Graph results are not shown in %.
Response: We appreciated the reviewer's comment. Therefore, modified it in all figure legends.
Line 203, RT-PCR should be corrected to qRT-PCR.
Response: We appreciated the reviewer's comment. Therefore, change it.
Line 211-212, Did you use a chelating agent (TPEN) in the experiments in this figure?
Response: We appreciated the reviewer's comment. Therefore, modified it.
Line 222-224 and 249-250, Please provide appropriate citations (used in the discussion).
Response: We appreciated the reviewer's comment. Therefore, changed the appropriate citations
Line 280-281, Is there a significant difference in Bcl-2 gene expression levels between Zn- and Zn+?
Response: We appreciated the reviewer's comment. Therefore, we removed it.
354-355, These are not discussion, but the results of Fig.4B.
Response: We appreciated the reviewer's comment. Therefore, added the results of Fig.4B.
Line 365-366, Why do we need a line break here?
Response: We appreciated the reviewer's comment. Therefore, modified it.

Reviewer 2 Report
In their manuscript, Lee et al addressed a question how zinc deficiency or supplementation affects leptin signaling and regulation in osteoblastic cells. Since leptin is a hormone that is largely secreted by adipocytes, the rationale of using a pre-osteoblastic cell line (MC3T3) to analyze the regulation of leptin by zinc is unclear. The physiological and/or therapeutical relevance of the findings of the study is poorly discussed, too. In general, the authors are kindly recommended to describe their findings more accurately and to refrain from stating more than their data actually show. I have a few further comments and suggest some additional experiments to improve the manuscript:
1). In fact, osteoblasts originate from mesenchymal stem cells, and osteocytes develop from the hematopoietic stem cell niche. This is, however, opposite to a sentence in the introduction (lines 37-39) – please correct.
2) The link between Zn supply, leptin signaling, apoptosis and osteoblast differentiation is poorly explained and discussed. How does Zn (absent or at low concentrations) cause leptin secretion in MC3T3-derived osteoblasts? Is release of leptin to the medium under Zn deficiency a result of apoptotic cell death? Do the authors propose adjusting a widely used protocol for MC3T3 differentiation in osteoblasts because supplementation of osteogenic media with Zn enhances osteoblastogenesis in vitro? Do the authors propose leptin as a paracrine regulator in osteoblasts?
3) Some key experimental procedures are poorly or unclearly described, e.g. cell treatments; qPCR procedure and mRNA expression calculation; band quantification by densitometry analysis. More specifically, “The cells were then treated with either 1 μM (Zn-deficient; Zn-) or 15 μM (Zn-adequate; Zn+) of ZnCl2 along with 5 μM of intracellular Zn chelator (TPEN) to induce the cellular Zn depletion” – does it mean that the cells were treated with 15 µM Zn + TPEN? I believe, treatment with 1µM Zn (‘low’ Zinc), 15µM Zn (‘high’ Zn) and TPEN alone (no Zn) would be more reasonable. Besides, I find labeling with “Zn-“ and “Zn+“ confusing as it should imply the supplementation of culture medium with a defined concentration of Zn (1µM or 15µM). Besides, I advocate for revising or clarifying the terms “Zn deficiency” (low Zn?) and “Zn deprivation” (no Zn? TPEN-treated?) because their usage in the text is also confusing
4). Please explain why treatment with “0 μM Zn” (i.e. no treatment?) results in a ~2-fold drop in cell viability (Fig. 1A) and in a ~7-fold increase in leptin release (Fig. 2A). What is actually the difference between the “0 μM Zn” and the “OSM” treatment? If “0 μM Zn” means TPEN treatment, please label accordingly.
5) Comments to Fig. 2:
i) How could the authors explain a ~4-fold increase of leptin secretion upon treatment with 1 μM Zn while its mRNA expression increased only ~1.5 times (Figs. 2A and 2B)?
ii) The fold-change difference in Fig. 2B (lower panel) does not correspond to the band intensities in the upper panel. Please explain how mRNA expression in Fig. 2B was calculated. Samples of what kind (cDNA? mRNA?) are shown in Fig. 2B (upper panel) as “Leptin mRNA”?
iii) By the appearance of bands in Fig. 2B (upper panel) it seems that the absolute expression of GAPDH is comparable or even higher than that of leptin in the same sample. Please, provide Ct values for both genes.
6). Samples of what kind (cDNA? mRNA?) are presented in Fig. 3A as “OB-Rb mRNA”? What is a possible reason that a band at “Zn-“ (Fig. 3A, upper panel) appears to migrate more slowly compared to the bands at two other conditions? mRNA expression at “Zn+” in Fig. 3A (lower panel) is comparable to that at “OSM”, but this is hardly reflected in the upper panel – please check your calculations and/or provide a representative gel image.
7) Comments to Fig. 4:
i) Although a phosphorylated form of STAT3 is known to localize to the nucleus and regulate gene expression, the authors have not analyzed its nuclear localization in their study. Therefore, it is quite an exaggeration to label the p-STAT3 bands with “nucleus” in Fig. 4B. For the same reason, the authors are recommended to refrain from stating “…our results suggested p-STAT3 protein was increased in the nucleus of osteoblastic MC3T3-E1 cells in Zn deficiency” (line 354f).
ii) Calculations of relative protein expression shown in the lower panels of the figure do not correspond to the intensities of bands shown in the upper panels – please check. Besides, the authors are suggested to calculate the amounts of phosphorylated forms of JAK2 and STAT3 relative to their unphosphorylated forms, rather than relative to GAPDH.
8) Comments to Fig. 5:
i) The authors are requested to show enlarged TEM images and to clearly mark nuclei and mitochondria in all images
ii) Based on what do the authors infer that it is mitochondria that are vacuolized? Is the mitochondrial function (respiration, membrane potential etc) also impaired in cells treated with 1 μM Zn?
iii) The authors are strongly recommended to confirm apoptosis and nuclear condensation in 1 μM-treated cells by an alternative method (e.g. Hoechst staining)
9) Induction of caspase-dependent apoptosis in cells treated with 1 μM Zn should be further confirmed, e.g. by analyzing the cleavage of a caspase-3 substrate, PARP1. In parallel, please also include a treatment with TPEN (to deprive cells of Zn).
10) Although the authors mention in Material & Methods that the cells were also treated with a Zn chelator (TPEN), most of the data are shown for the cells treated with either low (1 μM) or high (15 μM) concentration of Zn. It is suggested that the data with cells deprived of Zn (i.e. TPEN-treated) are shown in all figures as well, in parallel to Zn-treated cells.
11) I find the scheme in Fig. 7 too speculative and insufficiently supported by the data.
i) The authors highlight regulation of OB-R expression by STAT3 as “confirmed in this study”. There is no indication in literature that OB-R is regulated by STAT3, nor has this study provided adequate experimental evidence to confirm this.
ii) How can OB-R regulate the expression of leptin (indicated as “confirmed in this study”)? In this study, it is only shown that an increased expression of Ob-Rb gene goes along with an increased expression of leptin upon treatment with 1 μM Zn, which hardly indicates a direct interaction between the two genes.
iii) The statements “In response to leptin stimulation in the extracellular matrix of Zn deprived cells, the mRNA and protein expression of leptin receptor (OB-Rb) was increased via the activation of JAK2/STAT3 signaling pathway” and “…leptin production is induced by Zn deficiency upon the activation of the leptin signaling pathway” are not quite adequate to the data shown. In order to prove that it is leptin that causes the depicted downstream events and the upregulation of the OB-R expression in MC3T3 cells, this should be addressed in additional experiments where cells are treated with various concentrations of mouse leptin, in the absence/presence of Zn.
Other comments:
- The title of the manuscript is suggested to be revised because the intracellular concentration of Zn was not directly addressed in this study
- References #5 and #6 have little to do with the statement “It is widely known that Zn is associated with critical bone functions including growth, development, and maintenance of healthy bones in humans and other mammals” (lines 42-44). The authors should refer to the studies that are more specific to the role of zinc in the bone (not the role of zinc in general) and are thus more suitable to serve as references
- The letters “a-b-c” that are used in most of the plots are confusing. Please explain what they mean
- Please add kDa- and bp-markers to all gels/blots
- Please check suitability of all references (e.g. #10 in lines 46-47, #8 & #9 in lines 44-46)
- Line 34: “vitamin D” needs to be omitted from the list as it is apparently not a mineral
- How mRNA expression was calculated is not explained (ddCt method?). Besides, the sentence “To determine relative mRNA expression, housekeeping gene (GAPDH) and apoptosis marker gene with SYBR green I (SYBR Advantage qPCR Premix) were used” is misleading or likely wrong
- Line 250: please add a reference
- Line 269f: “Deficiency of Zn resulting osteoporosis through the loss of bone mass has been extensively studied”. Although “extensively studied”, the authors support this statement with only one reference, which is one of their previous studies
- Line 322f “…the pleotropic hormone leptin has now been reported to modulate bone mineral density …” – please omit “now” or refer to a paper that is more recent than from the year 2000
- Please refer to more recent publications where applicable
Author Response
Response to Reviewer
We thank you for giving the opportunity to submit a revised and updated manuscript of “Depletion of (intracellular) zinc causes osteoblast apoptosis with elevation of leptin secretion of phosphorylation of JAK2/STAT3”. We deeply appreciate your time and expertise that you dedicated for providing feedback on our manuscript. We are grateful for the productive and insightful comments on and valuable improvements to our paper.
We have modified, corrected the manuscript and incorporated most of the comments and suggestions that you suggested as much as we can. Please also see below, for a point-by-point response to the reviewer’s comments and concerns. Again, the authors deeply appreciate your constructive comments!
In their manuscript, Lee et al addressed a question how zinc deficiency or supplementation affects leptin signaling and regulation in osteoblastic cells. Since leptin is a hormone that is largely secreted by adipocytes, the rationale of using a pre-osteoblastic cell line (MC3T3) to analyze the regulation of leptin by zinc is unclear. The physiological and/or therapeutical relevance of the findings of the study is poorly discussed, too. In general, the authors are kindly recommended to describe their findings more accurately and to refrain from stating more than their data actually show. I have a few further comments and suggest some additional experiments to improve the manuscript:
Response: Leptin is largely secreted by adipocytes in general. In addition that, the emerging studies reported that leptin and leptin receptors are also expressed in osteoblasts too (Roseland et al, 2001; Lee et al, 2002). The issues for leptin secretion and regulation in osteoblasts and bone are still controversial to be elucidated (Zeadin et al, 2018)
As the reviewer suggested, we described our findings more accurately and stated more than our data actually show as much as we can
1) In fact, osteoblasts originate from mesenchymal stem cells, and osteocytes develop from the hematopoietic stem cell niche. This is, however, opposite to a sentence in the introduction (lines 37-39) – please correct.
Response: We fully understand and agree with the reviewer’s comments. Therefore, we modified the introduction.
2) The link between Zn supply, leptin signaling, apoptosis and osteoblast differentiation is poorly explained and discussed. How does Zn (absent or at low concentrations) cause leptin secretion in MC3T3-derived osteoblasts? Is release of leptin to the medium under Zn deficiency a result of apoptotic cell death? Do the authors propose adjusting a widely used protocol for MC3T3 differentiation in osteoblasts because supplementation of osteogenic media with Zn enhances osteoblastogenesis in vitro? Do the authors propose leptin as a paracrine regulator in osteoblasts?
Response: We greatly appreciate the positive comments by the reviewer. We also understand the reviewer’s expert comment about the relation of Zn and leptin. Leptin was initially identified to be secreted by fat tissues or adipocytes. There are reports on bone cell types producing substantial amounts of leptin which includes skeletal muscle, bone marrow, primary cultures of osteoblasts (Reseland et al, 2001) as well as cultures of MC3T3-E1 cells (Kume et al, 2002). Different types of cells were analyzed by Western blot for leptin expression. Leptin protein was expressed in osteoblastic MC3T3-E1 cells under normal condition (Figure 1). However, these preliminary results are shown only to the reviewer for reviewing purpose.
Figure 1. Leptin protein expression in tissues, epithelial cells and osteoblastic MC3T3-E1 cells. MC3T3-E1 cells were cultured in normal osteogenic medium (α-MEM with 10% FBS, 1 mM sodium pyruvate, 100 units/ml penicillin and 100 ug/ml streptomycin) plus 3 mM monosodium phosphate (inorganic phosphate) and 50 ug/ml L-ascorbic acid. EA.hy 926 cells were cultured in DMEM with 10% FBS, 2% HAT, 100 units/ml penicillin and 100 ug/ml streptomycin). Tissue samples were obtained from rats fed with Zn-adequate diet (35 mg Zn/kg diet).
3) Some key experimental procedures are poorly or unclearly described, e.g. cell treatments; qPCR procedure and mRNA expression calculation; band quantification by densitometry analysis. More specifically, “The cells were then treated with either 1 μM (Zn-deficient; Zn-) or 15 μM (Zn-adequate; Zn+) of ZnCl2 along with 5 μM of intracellular Zn chelator (TPEN) to induce the cellular Zn depletion” – does it mean that the cells were treated with 15 µM Zn + TPEN? I believe, treatment with 1µM Zn (‘low’ Zinc), 15µM Zn (‘high’ Zn) and TPEN alone (no Zn) would be more reasonable. Besides, I find labeling with “Zn-“ and “Zn+“ confusing as it should imply the supplementation of culture medium with a defined concentration of Zn (1µM or 15µM). Besides, I advocate for revising or clarifying the terms “Zn deficiency” (low Zn?) and “Zn deprivation” (no Zn? TPEN-treated?) because their usage in the text is also confusing
Response: Apologize for the confusion. OSM is the ‘normal osteogenic differentiation medium’ without Zn chelator, TPEN. 1 μM (low Zn) and 15 μM (high Zn) are the media Zn added as designated with TPEN. Revised on the manuscript.
4). Please explain why treatment with “0 μM Zn” (i.e. no treatment?) results in a ~2-fold drop in cell viability (Fig. 1A) and in a ~7-fold increase in leptin release (Fig. 2A). What is actually the difference between the “0 μM Zn” and the “OSM” treatment? If “0 μM Zn” means TPEN treatment, please label accordingly.
Response: We fully understand the reviewer’s expert comments. We honestly do not know the reason(s) why this happened. One possible explanation is Fig. 1A is the result of the osteoblast viability itself, while Fig 1A is the result of leptin concentration in medium which is the osteoblast secreted, therefore can be accumulated leptin levels by Zn.
5) Comments to Fig. 2:
- i) How could the authors explain a ~4-fold increase of leptin secretion upon treatment with 1 μM Zn while its mRNA expression increased only ~1.5 times (Figs. 2A and 2B)?
Response: We fully understand the reviewer’s expert comments. As the same potential explanation above, the result of ~4-fold increase of leptin secretion upon treatment with 1 μM Zn (Fig. 2A) was measured in medium, while its mRNA expression ~1.5 fold increase was in cells (Figs. 2A and 2B), which may not be consistent in intensity.
- ii) The fold-change difference in Fig. 2B (lower panel) does not correspond to the band intensities in the upper panel. Please explain how mRNA expression in Fig. 2B was calculated. Samples of what kind (cDNA? mRNA?) are shown in Fig. 2B (upper panel) as “Leptin mRNA”?
Response: We fully understand the reviewer’s expert comments. We honestly do not know the reason(s) why this happened. One possible explanation is Fig. 1A is the result of the osteoblast viability itself, while Fig 1A is the result of leptin concentration in medium which is the osteoblast secreted, therefore can be accumulated leptin levels by Zn.
iii) By the appearance of bands in Fig. 2B (upper panel) it seems that the absolute expression of GAPDH is comparable or even higher than that of leptin in the same sample. Please, provide Ct values for both genes.
Response: We appreciate your comments. Honestly, we used the RT-PCR analysis. Therefore, we added method and modified manuscript.
6). Samples of what kind (cDNA? mRNA?) are presented in Fig. 3A as “OB-Rb mRNA”? What is a possible reason that a band at “Zn-“ (Fig. 3A, upper panel) appears to migrate more slowly compared to the bands at two other conditions? mRNA expression at “Zn+” in Fig. 3A (lower panel) is comparable to that at “OSM”, but this is hardly reflected in the upper panel – please check your calculations and/or provide a representative gel image.
Response: “OB-Rb mRNA means the replication of cDNA. About the possible reason, ‘why a band at “Zn-“ (Fig. 3A, upper panel) appears to migrate more slowly compared to the bands at two other OSM and Zn+ conditions, we still cannot identify about it unfortunately. Authors appreciate again for reviewer expert comment.
7) Comments to Fig. 4:
- i) Although a phosphorylated form of STAT3 is known to localize to the nucleus and regulate gene expression, the authors have not analyzed its nuclear localization in their study. Therefore, it is quite an exaggeration to label the p-STAT3 bands with “nucleus” in Fig. 4B. For the same reason, the authors are recommended to refrain from stating “…our results suggested p-STAT3 protein was increased in the nucleus of osteoblastic MC3T3-E1 cells in Zn deficiency” (line 354f).
Response: Unfortunately we couldn’t analyze STAT2 nuclear localization due to sample in shortage under the same experimental condition. As the reviewer suggested, we refrain the word ‘nucleus’ in the text as well as in the figure and revised the manuscript.
- ii) Calculations of relative protein expression shown in the lower panels of the figure do not correspond to the intensities of bands shown in the upper panels – please check. Besides, the authors are suggested to calculate the amounts of phosphorylated forms of JAK2 and STAT3 relative to their unphosphorylated forms, rather than relative to GAPDH.
Response: 1) We checked the relative protein expression (Fig. 4A, low panel) and the band images (Fig. 4B. upper panel) and revised it, as the reviewer suggested. 2) For the calculation of the amounts of phosphorylated forms of JAK2 and STAT3 relative to their unphosphorylated forms are considered as the reviewer suggested.
8) Comments to Fig. 5:
- i) The authors are requested to show enlarged TEM images and to clearly mark nuclei and mitochondria in all images.
Response: We understand the reviewer’s excellent question. Unfortunately, we do not have enlarged TEM images.
- ii) Based on what do the authors infer that it is mitochondria that are vacuolized? Is the mitochondrial function (respiration, membrane potential etc) also impaired in cells treated with 1 μM Zn?
Response: We understand the reviewer’s excellent question. As our understanding, mitochondria are vacuolized too as the sign of apoptosis examined by TEM. Unfortunately we couldn’t confirm the various mitochondrial function (respieration, membrane potential etc.) under this experimental condition.
iii) The authors are strongly recommended to confirm apoptosis and nuclear condensation in 1 μM-treated cells by an alternative method (e.g. Hoechst staining)
Response: We appreciate the reviewer’s excellent suggestion. Therefore, we evaluated apoptosis and nuclear condensation in 1 μM-treated cells by Hoechst staining in new Figure 6.
9) Induction of caspase-dependent apoptosis in cells treated with 1 μM Zn should be further confirmed, e.g. by analyzing the cleavage of a caspase-3 substrate, PARP1. In parallel, please also include a treatment with TPEN (to deprive cells of Zn).
Response: We understand the reviewer’s instructive question. To respond to the reviewer’s comment, we have also performed additional experiments in new Figure 7.
10) Although the authors mention in Material & Methods that the cells were also treated with a Zn chelator (TPEN), most of the data are shown for the cells treated with either low (1 μM) or high (15 μM) concentration of Zn. It is suggested that the data with cells deprived of Zn (i.e. TPEN-treated) are shown in all figures as well, in parallel to Zn-treated cells.
Response: Apologize for the confusion. OSM is the ‘normal osteogenic differentiation medium’ without Zn chelator, TPEN. 1 μM (low Zn) and 15 μM (high Zn) are the media Zn added as designated with TPEN. Revised on the manuscript.
11) I find the scheme in Fig. 7 too speculative and insufficiently supported by the data.
- i) The authors highlight regulation of OB-R expression by STAT3 as “confirmed in this study”. There is no indication in literature that OB-R is regulated by STAT3, nor has this study provided adequate experimental evidence to confirm this.
- ii) How can OB-R regulate the expression of leptin (indicated as “confirmed in this study”)? In this study, it is only shown that an increased expression of Ob-Rb gene goes along with an increased expression of leptin upon treatment with 1 μM Zn, which hardly indicates a direct interaction between the two genes.
Response: As the reviewer bring about, this study couldn’t provide direct evidence to confirm that OB-R is regulated by STAT3, but we observed and presented an increased expression of Ob-Rb gene along with the increased expression of leptin under the Zn deficiency (treatment with 1 μM Zn) in osteoblasts. We deleted the scheme in Fig. 7 as your excellent comments.
iii) The statements “In response to leptin stimulation in the extracellular matrix of Zn deprived cells, the mRNA and protein expression of leptin receptor (OB-Rb) was increased via the activation of JAK2/STAT3 signaling pathway” and “…leptin production is induced by Zn deficiency upon the activation of the leptin signaling pathway” are not quite adequate to the data shown. In order to prove that it is leptin that causes the depicted downstream events and the upregulation of the OB-R expression in MC3T3 cells, this should be addressed in additional experiments where cells are treated with various concentrations of mouse leptin, in the absence/presence of Zn.
Response: We understand the reviewer’s excellent question. Unfortunately, we do not have the enough cell sample leftover and time to do these experiments. We will do future experiments.
Other comments:
- The title of the manuscript is suggested to be revised because the intracellular concentration of Zn was not directly addressed in this study.
Response: We fully understand and agree with the reviewer’s comments. Therefore, we modified the title.
- References #5 and #6 have little to do with the statement “It is widely known that Zn is associated with critical bone functions including growth, development, and maintenance of healthy bones in humans and other mammals” (lines 42-44). The authors should refer to the studies that are more specific to the role of zinc in the bone (not the role of zinc in general) and are thus more suitable to serve as references
Response: We fully understand and agree with the reviewer’s comments. Therefore, we changed the references.
- The letters “a-b-c” that are used in most of the plots are confusing. Please explain what they mean
Response: We fully understand and agree with the reviewer’s comments. Therefore, we added the description in figure legends.
- Please add kDa- and bp-markers to all gels/blots
Response: We appreciate the reviewer’s expert points. We added it.
- Please check suitability of all references (e.g. #10 in lines 46-47, #8 & #9 in lines 44-46)
Response: We appreciate the reviewer’s expert points. We checked it.
- Line 34: “vitamin D” needs to be omitted from the list as it is apparently not a mineral
Response: We appreciate the reviewer’s expert points. We deleted it.
- How mRNA expression was calculated is not explained (ddCt method?). Besides, the sentence “To determine relative mRNA expression, housekeeping gene (GAPDH) and apoptosis marker gene with SYBR green I (SYBR Advantage qPCR Premix) were used” is misleading or likely wrong
Response: We appreciate the reviewer’s expert points. We modified it.
- Line 250: please add a reference
Response: We appreciate the reviewer’s expert points. We added it.
- Line 269f: “Deficiency of Zn resulting osteoporosis through the loss of bone mass has been extensively studied”. Although “extensively studied”, the authors support this statement with only one reference, which is one of their previous studies
Response: We appreciate the reviewer’s expert points. We modified it.
- Line 322f “…the pleotropic hormone leptin has now been reported to modulate bone mineral density …” – please omit “now” or refer to a paper that is more recent than from the year 2000
Response: We appreciate the reviewer’s expert points. Unfortunately, we could not find the recent reports about zinc and leptin studies.
- Please refer to more recent publications where applicable
Response: We appreciate the reviewer’s expert points. Unfortunately, we could not find the recent reports the most properly, however as it becomes available we refer and update them.

Round 2
Reviewer 1 Report
I have verified that the corrections were properly made.
Author Response
Once again, thank yo very much for your comments. The reviewer comments are meaningful and helpful to improve the manuscript.
Reviewer 2 Report
In the revised version of the manuscript, the authors have included data of some additional experiments that were suggested. However, the following previously raised comments and suggestions remained insufficiently or improperly addressed, namely:
- a wrong statement about the origin of osteoblasts and osteoclasts has not been corrected in the introduction (line 36-38)
- that it is known that leptin itself is expressed in osteoblasts has not been added to the main text and/or introduction (although claimed in the authors’ response), thus leaving the rationale of the study obscure
- the discussion has not been improved, while some critical questions remained unanswered or poorly explained in the authors’ response and in the main text (e.g. about the mechanistic link between Zn, leptin signaling and osteoblast differentiation). The message and the significance of the study are still hardly clear to the reader
- the authors used RT-PCR instead of qRT-PCR to estimate gene expression but have not provided the number of cycles used, which makes the result hardly reliable
- in the revised version of the Material & Methods (lines 136-138) the authors claim to have applied cytoplasm/nuclear fractionation for Fig. 4. However, the Fig. 4A looks like either the fractionation was unsuccessful (as both the cytoplasmic and the nuclear marker could be detected in the same sample), or the presentation of the data is simply inappropriate and confusing
- the results of densitometry analyses in some cases apparently do not match the band intensities shown in blots/gels (e.g. Fig. 2B, 3A, 3B)
- bigger (or original) TEM images were not provided, though requested. Lack of clearer images makes it impossible for the reader to evaluate the data and thus lowers the manuscript quality
- the authors show no PARP1 cleavage under “low Zn” (only the full-length form of PARP1 is shown in Fig.7A), which does not corroborate the authors' idea that caspase-dependent apoptosis is triggered. Nor have these newly obtained data been embeded and discussed in the authors’ response or in the manuscript
- a poor (or poorly explained) rationale of adding a Zn-chelator (TPEN) together with Zn to the culture medium is, in my opinion, the main drawback of the study and its research design. If the effect of extracellularly added Zn is to be studied while intracellular Zn is depleted, TPEN should be added before Zn treatment, but not together with Zn - in order to exclude chelation of the extracellular Zn with TPEN. Again, the authors have not clarified this question in the manuscript and their response, and this experimental procedure is still ambiguously described.
Author Response
(authors response)
Again, we authors would like to thank Reviewer for taking the necessary time and effort to review the manuscript. We sincerely appreciate all your valuable comments and suggestions below , which helped us in improving the quality of the manuscript.
In the revised version of the manuscript, the authors have included data of some additional experiments that were suggested. However, the following previously raised comments and suggestions remained insufficiently or improperly addressed, namely:
- a wrong statement about the origin of osteoblasts and osteoclasts has not been corrected in the introduction (line 36-38)
Response: We fully understand and agree with the reviewer’s comments. Therefore, we modified in the introduction.
- that it is known that leptin itself is expressed in osteoblasts has not been added to the main text and/or introduction (although claimed in the authors’ response), thus leaving the rationale of the study obscure
Response: We fully understand and agree with the reviewer’s comments. Therefore, we added the rationale of the study.
- the discussion has not been improved, while some critical questions remained unanswered or poorly explained in the authors’ response and in the main text (e.g. about the mechanistic link between Zn, leptin signaling and osteoblast differentiation). The message and the significance of the study are still hardly clear to the reader
Response: We appreciate your comments. Therefore, we modified in the discussion.
- the authors used RT-PCR instead of qRT-PCR to estimate gene expression but have not provided the number of cycles used, which makes the result hardly reliable
Response: We appreciate the reviewer’s excellent suggestion. Therefore, we added the number of cycles used.
- in the revised version of the Material & Methods (lines 136-138) the authors claim to have applied cytoplasm/nuclear fractionation for Fig. 4. However, the Fig. 4A looks like either the fractionation was unsuccessful (as both the cytoplasmic and the nuclear marker could be detected in the same sample), or the presentation of the data is simply inappropriate and confusing
Response: We appreciate your comments. Therefore, we deleted the Fig. 4A and modified the methods.
- the results of densitometry analyses in some cases apparently do not match the band intensities shown in blots/gels (e.g. Fig. 2B, 3A, 3B)
Response: We appreciate the reviewer’s expert points. We checked the band intensities and modified it.
- bigger (or original) TEM images were not provided, though requested. Lack of clearer images makes it impossible for the reader to evaluate the data and thus lowers the manuscript quality
Response: We understand the reviewer’s instructive question. To respond to the reviewer’s comment, we changed the bigger TEM image.
- the authors show no PARP1 cleavage under “low Zn” (only the full-length form of PARP1 is shown in Fig.7A), which does not corroborate the authors' idea that caspase-dependent apoptosis is triggered. Nor have these newly obtained data been embeded and discussed in the authors’ response or in the manuscript
Response: We appreciate the reviewer’s expert points. We added the PARP1 cleavage form and modified it.
- a poor (or poorly explained) rationale of adding a Zn-chelator (TPEN) together with Zn to the culture medium is, in my opinion, the main drawback of the study and its research design.
If the effect of extracellularly added Zn is to be studied while intracellular Zn is depleted, TPEN should be added before Zn treatment, but not together with Zn - in order to exclude chelation of the extracellular Zn with TPEN.
Again, the authors have not clarified this question in the manuscript and their response, and this experimental procedure is still ambiguously described.
Response: TPEN is an intercellular membrane-permeable ion chelator which has particularly high affinity for zinc. It is a cell-permeable zinc chelator (1-3). Since TPEN is cell memebrance-permeable, it can deplete zinc extracellularly as well as intracellularly therefore a more strict zinc chelator, compare to extracellular zinc chelexing agent such as Chelex resin 100 (Bio-Rad) (4) etc.
The experimental design of for zinc treatment in this study was depelting in and outside zinc using intracellular zinc chelator TPEN which gives the same baseline for cellular zinc depletion and at the same time zinc added along with the TPEN within the range of zinc (0-15 micorM) which cover from physiologically low to adequate/sufficient zinc level. If TPEN is added before zinc treatment, the cells would be in apoptosis which is not under the physiological condition for the experiment. The study aim is not particularly to examine the effect of 'extracellular added Zn', but physiologically low to high zinc effect in osteoblasts.
The reviewer comments are meaningful and helpful to improve the manuscript.
References:
- TPEN (catalog #: 4309). https://www.rndsystems.com/products/tpen_4309, https://www.tocris.com/products/tpen_4309
- Cho YE, Lomeda RR. Ryu SH, Lee JH, Beattie JH, Kwun IS (2007). Cellular Zn depletion by metal ion chelators (TPEN, DTPA and chelex resin) and its application to osteoblastic MC3T3-E1 cells. Nutrition Research and Practice. 1(1): 29-35. doi : 10.4162/nrp.2007.1.1.29 . PMC 2882573 . PMID 20535382 .
- He S, Zou Y, Zhan M, Guo Q, Zhang Y, Zhang Z, Li B, Zhang S, Chu H (2020). Zinc Chelator N,N,N′,N′-Tetrakis(2-Pyridylmethyl)Ethylenediamine Reduces the Resistance of Mycobacterium abscessus to Imipenem. Intection and Drug Resistance 13:2883-289. https://iovs.arvojournals.org/article.aspx?articleid=2123166
- Herui Wang, Bin Li, Kulsum Asha, Ryan L. Pangilinan, Asha Thuraisamy, Harman Chopra, Susumu Rokudai ,Yong Yu1, Carol L. Prives, and Yan Zhu1 (2021). The ion channel TRPM7 regulates zinc-depletion-inducedMDMX degradation. J Biol. Chem. 297(5):101292- https://doi.org/10.1016/j.jbc.2021.101292
https://www.jbc.org/article/S0021-9258(21)01097-8/pdf
